# Augmented Reality-Based Explainable AI Strategies for Establishing Appropriate Reliance and Trust in Human-Robot Teaming

Matthew B. Luebbers*
*Department of Computer Science*
*University of Colorado, Boulder*
Boulder, Colorado, USA
matthew.luebbers@colorado.edu

Aaquib Tabrez*
*Department of Computer Science*
*University of Colorado, Boulder*
Boulder, Colorado, USA
mohd.tabrez@colorado.edu

Bradley Hayes
*Department of Computer Science*
*University of Colorado, Boulder*
Boulder, Colorado, USA
bradley.hayes@colorado.edu

*Abstract*—In human-robot teaming, live and effective communication is of critical importance for maintaining coordination and improving task fluency, especially in uncertain environments. Poor communication between teammates can foster doubt and misunderstanding, and lead to task failures. In previous work, we explored the idea of visually communicating notions of environmental uncertainty alongside robot-generated suggestions through augmented reality (AR) interfaces in a human-robot teaming setting. We introduced two complementary modalities of visual guidance: prescriptive guidance (visualizing recommended actions), and descriptive guidance (visualizing state space information to aid in decision-making), along with an algorithm to generate and utilize these modalities in partially-observable multi-agent collaborative tasks. We compared these modalities in a human subjects study, where we showed the ability of this combined guidance to improve trust, interpretability, performance, and human teammate independence. In this new work, we synthesize key takeaways from that study, leveraging them to describe remaining open challenges for live communication for human-robot teaming under uncertainty, and propose a set of approaches to address them via a collection of explainable AI techniques such as visual counterfactual explanations, predictable and explicable planning, and robot-generated justifications.

*Index Terms*—Human-Robot Collaboration, Explainable AI, Augmented Reality, Reinforcement Learning, Counterfactual Explanation, Shared Mental Models, Plan Justification

## I. INTRODUCTION & MOTIVATION

In tasks involving teamwork across uncertain, changing environments, communication quality can be the difference between success and failure. This is especially relevant for human-robot teams, where team members do not share any inherent common basis for communication. Autonomous agents are well-equipped to operate in probabilistic state spaces, gathering observations and choosing optimal actions in response to new information. In order to contribute to enhanced team performance however, we posit that these agents should communicate this knowledge to human teammates, allowing for a coordinated problem-solving strategy.

This work was funded as part of the Army Research Lab STRONG Program (#W911NF-20-2-0083).
*These authors contributed equally to this work.

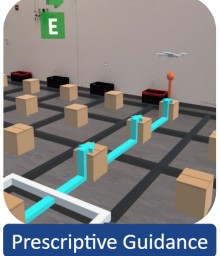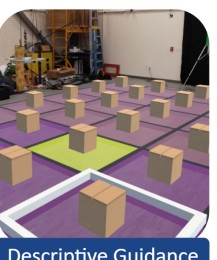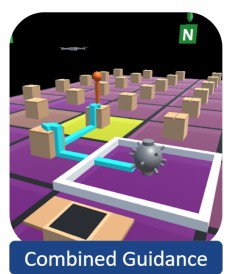

Fig. 1. AR-based visual guidance modalities: prescriptive guidance, featuring arrows and pins (left), descriptive guidance, featuring an environmental heatmap (middle), and a combination of both guidance types (right) in an experimental Minesweeper-inspired domain.

Consider a search task, where human and robot teammates spread out to locate targets over a large environment. This domain would be particularly well-suited to visual communication via augmented reality (AR) headset. Visual modalities are well suited to convey pieces of information which involve uncertainty or noise [7], and AR-based head-mounted interfaces in particular have ergonomic advantages due to their hands-free nature [16], allowing for rapid communication with no need for user context switching [20], ideal for tasks where humans need to traverse terrain under time pressure. What's more, AR's ability to visualize in-situ meshes well with the concept of explainable AI (xAI); multiple works have utilized AR visualizations to illuminate the otherwise opaque planning processes of both mobile robots and robotic manufacturing arms in various teaming scenarios [25], [31], [39]. Within xAI, visualization is a common modality for presenting explanations to expose overconfidence in models [1], visualize class boundaries [30], and aid AI experts in debugging [19]. However, most of these techniques assume a degree of AI expertise combined with domain-specific knowledge [28], [29]; our aim was to leverage AR to design user-friendly visual communication interfaces usable in live human-robot teaming scenarios.

We explored the use of AR-based visual guidance for human-robot teaming in our prior work introducing MARS (Min-entropy Algorithm for Robot-supplied Suggestions) [35].

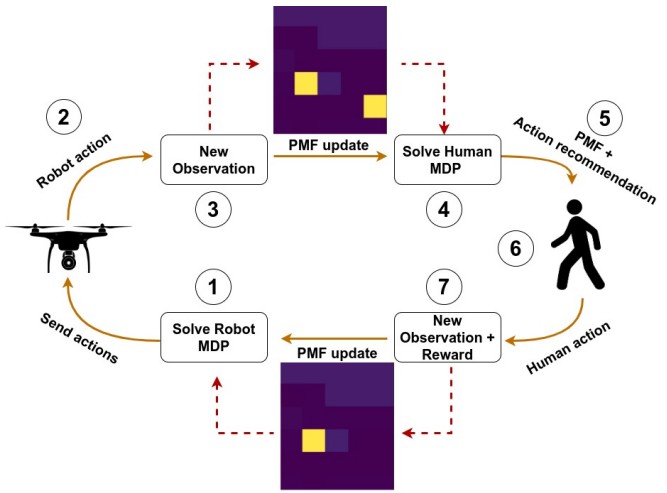

Fig. 2. Flowchart of the MARS algorithm's operation in a stochastic collaborative task with human and robotic teammates.

MARS consists of both an algorithm for multi-agent collaboration under uncertainty, as well as a collection of AR guidance interfaces for human teammates generated by that algorithm, compared and contrasted via user study. In this paper, we first summarize our findings from MARS regarding the design of visual guidance systems. We then describe a set of remaining open challenges for communication in human-robot teaming, along with proposed approaches to address them.

## II. SUMMARY OF MARS

In our prior work [35], we first introduced our planning algorithm for uncertain environments, informing the generation of proactive visual recommendations. We also provided characterizations of two complementary modalities of visual guidance: prescriptive guidance (visualizing recommended actions), and descriptive guidance (visualizing state space information to aid in decision-making). We evaluated the effectiveness of our algorithm and compared these different modalities of AR-based guidance in a human subjects study involving a collaborative, partially observable search task.

### A. Algorithm

The core insight behind our algorithm is that environmental uncertainty over task-relevant variables can be succinctly characterized by a dynamically-updating probability mass function (PMF), a common practice across a variety of search tasks [13], [40], [41]. That PMF can serve as a shared utility function common to all agents (both human and autonomous), and can be communicated to human teammates as it changes in response to autonomous agent observations, providing insight into the agent's policy. This PMF is utilized by two separate Markov Decision Processes (MDPs); one for controlling autonomous agents ($M_R$) designed to minimize entropy across the environment, and another for generating assistive guidance for the human teammate ($M_H$) aimed at minimizing search time. We solve both of these MDPs via online reinforcement learning to get optimal policies for autonomous agents and

action recommendations for human teammates respectively, continually updating the shared PMF as the basis for their reward functions.

As shown in Figure 2, we modeled the broader turn-taking interaction as follows: 1) the robot's MDP $M_R$ is solved, parametrized by the PMF, and actions are sent to all agents, 2) the robots take their actions, and 3) get new observations, updating the PMF. 4) the updated PMF is used to solve for a policy in the human's MDP $M_H$, and 5) the resulting PMF and action recommendations from that policy are communicated to the human, who 6) views the guidance via an AR interface and takes an action, 7) leading to a new observation and reward, updating the PMF once more. For more details on the algorithmic formulation of MARS, reference Section 3 of the original work [35].

### B. Prescriptive & Descriptive Visual Guidance

Central to the contribution of MARS is a characterization of two distinct visual guidance modalities, which were deployed as AR interfaces and correspond to the two data products of the algorithm. First is "prescriptive guidance", the essence of which is directly suggesting to a human teammate what actions they should take next. For example, in tasks involving physically navigating through space, movement suggestions can be represented as holographic arrows projected onto the ground, extending from the human's current location to their next suggested waypoint (Figure 1 Left). This type of guidance is designed to be straightforward and require little mental effort to follow. However, since the recommendations are presented sans rationale, they require a degree of human trust in the system.

The second modality is "descriptive guidance", which provides state space information for human teammates to use to inform their decision-making. For spatial navigation tasks, this takes the form of the PMF visualized as a heatmap, projected onto the environment itself, dividing the space into regions and coloring each square according to its expected reward (Figure 1 Center). In contrast to the more explicit prescriptive guidance, the descriptive modality acts as a decision support tool that empowers human teammates to plan by themselves, trading higher cognitive load for increased flexibility and transparency.

### C. Experiment

We evaluated the utility of our visual guidance modalities and the effectiveness of our algorithm through a human subjects study using a 3D turn-based collaborative analogue of the PC game Minesweeper, played using a HoloLens 2 AR headset (Figure 1). We tasked a team consisting of a human and a simulated drone with locating and defusing a number of mines hidden throughout a grid of cardboard boxes projected onto the floor of an experiment space. The drone is able to autonomously navigate the environment, taking measurements with a noisy sensor (false-positive rate: 10%, false-negative rate: 1%) to attempt to determine whether a box contains a hidden mine. The human must also physically navigate the

environment, spending turns to search boxes and defuse mines whenever they think they've located one.

We conducted a 3 × 1 within-subjects study with each condition varying the type of visual guidance given to the human teammate by the drone: 1) prescriptive guidance, or the 'arrow' condition, 2) descriptive guidance, or 'heatmap', and 3) a combination of prescriptive and descriptive guidance, or 'combined' (Figure 1).

### D. Key Results

Through our user study, we tested five key hypotheses (three subjective and two objective). The subjective hypotheses were as follows: participants will find the guidance in the 'combined' condition more trustworthy (H1.a), interpretable (H1.b), and less stressful (H1.c) compared to the other guidance types. H2 focused on performance, stating that participants will take less time to solve the task in the 'combined' or 'arrow' conditions compared with 'heatmap'. H3 states that participants will act with more independence and deviate more frequently from the prescribed path in the 'combined' condition compared with 'arrow'. To evaluate these hypotheses, we used a variety of subjective, survey-based measures along with several objective metrics (total moves, total time, time per move, and guidance compliance rate). Section 5 of the original work [35] gives more details as to the experiment setup and measures, while Section 6 describes the results and analysis.

We found statistically significant support for hypotheses H1.a (trustworthiness) and H1.b (interpretability) using subjective scales, while lacking enough evidence to validate H1.c (stress). We also validated H2 (performance) and H3 (independence) via objective metrics. Exit interviews from participants provided further evidence in support of the aforementioned hypotheses. In summary, we found that combining visual insights into environmental uncertainty (descriptive guidance) with robot-provided action suggestions (prescriptive guidance) improved trust, interpretability, and performance, and made human collaborators more independent.

## III. Open Challenges & Future Approaches

In this section, we summarize additional actionable takeaways from our user study, synthesizing them into overarching design goals for future visual guidance systems for human-robot teaming, as well as provide an outline of proposed approaches to address those goals, drawn from a collection of xAI and AR-based visualization techniques.

Firstly, although prescriptive guidance is simple to follow and frequently arises as a natural choice for navigation systems (e.g., getting directions via Google Maps), it puts the human user into a subconscious 'automatic' thinking pattern (also known as system 1 thinking [23]). This pattern is inherently restrictive, as it limits the user's ability to react and adapt to uncertain situations or sub-optimal system recommendations. Descriptive guidance, on the other hand, forces users into system 2 thinking, where they must actively plan ahead and calculate their next actions. By combining these two visual guidance approaches, human teammates have the capacity to

reduce their workload by leveraging the explicit prescriptive guidance, while maintaining environmental awareness and acting with greater independence when called for. However, prior research has shown that users of guidance systems will subconsciously shift towards system 1 thinking over time as they observe the system performing reliably [8], [22]. Therefore, our first overarching design goal is to leverage differing visual guidance types to properly calibrate system 1 and system 2 thinking in human teammates operating in uncertain environments.

Secondly, in the 'arrow' condition, participants' trust in system suggestions was highly inconsistent; some over-trusted the guidance (taking its suggestions to be inherently correct, leading to Type I errors [9]), while some under-trusted it (frequently ignoring the arrow in order to act more conservatively, leading to Type II errors). These wildly differing priors for perceived system reliability were highly dependent on participants' prior exposure to AI and robotics. Ideally, we would want human teammates to accept high-quality advice ('appropriate compliance') and reject low-quality advice ('appropriate reliance') [9], [14], which is challenging to achieve when providing explicit recommendations. Our study found evidence that presenting descriptive guidance alongside prescriptive led to user trust settling somewhere in the middle of the two extremes. This aligns with xAI findings suggesting that system interpretability mitigates over- and under-trust [6], [38], and that better understanding of system functioning increases acceptance of system recommendations [32], [42]. Our second overarching design goal relates to this issue of trust calibration; in stochastic environments with high uncertainty of recommendations, visual guidance systems should have the ability to communicate when advice is of high or low confidence to appropriately steer user trust.

Considering these two design goals for future visual guidance systems in human-robot teaming, we propose the following approaches:

**A1: Counterfactual Explanation.** Counterfactual explanation refers to a suite of techniques from xAI, where specific changes to the inputs of known models are identified that lead to changes in output classification in order to intuit a causal link [24], [27]. Counterfactual questions are generally posed with the form "Why did $P$ happen rather than $Q$?" For example, a counterfactual explanation in an image classifier might read "This is an image of an alligator rather than of a crocodile, because it has a rounded snout instead of a pointed snout." These techniques have the capacity to provide context about internal model reasoning to users, leading to usefulness for model debugging and failure recovery [4], [15], [26], [36].

Following our study, many participants noted that they found it difficult to notice changes to the descriptive guidance when the change did not occur in their field of view. These participants suggested adding a feature alerting users whenever the robot teammate discovers a new high confidence target, to enable more informed decision making. When this happens, it would be desirable to convey how this reward has changed, which can be achieved through the use of counterfactual

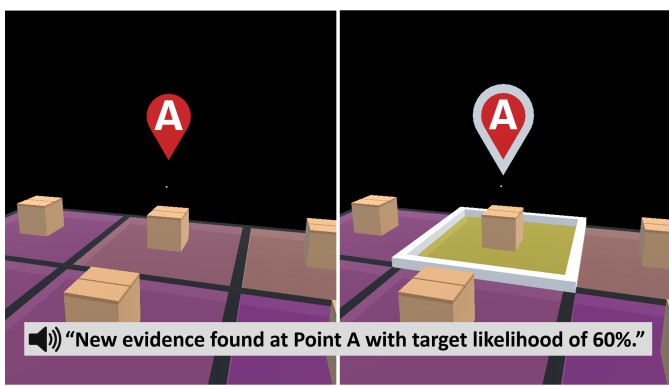

Fig. 3. Prototype counterfactual alert in the Minesweeper domain. The audio cue directs the user's attention to point A and states the new reward confidence, while the holographic visualization cycles between the left (old reward) and right (new reward) images.

techniques. We propose using a combination of visual and natural-language based counterfactuals to convey three pieces of information in the event of significant feature changes in the environment: 1) the spatial location of the reward change, 2) how the reward has changed, and 3) the system's confidence in the reward change. The first two components are designed to get users to pause and deliberate their plan, shifting from system 1 to system 2 thinking. The third component helps users assess the relevance of this new information and is aimed at mitigating over- and under-trust.

Figure 3 showcases a prototype system design for conveying counterfactual alerts to users in the Minesweeper domain. In response to a new drone observation, a landmark pin labeled A is introduced, indicating the spatial location of the reward change. AR-based visualization allows for the creation of features such as this out of thin air to quickly reference points in physical space that would otherwise be difficult to describe. To showcase how the reward has changed in response to a new drone observation, the AR visualization cycles for a period between Figure 3 Left (the old reward PMF value) and Figure 3 Right (the new reward PMF value, now highlighted in white). This cycling visualization is accompanied by a natural language audio cue describing the reward change and its associated confidence.

**A2: Minimizing Cognitive Load.** The next set of techniques are aimed at minimizing the cognitive load of human teammates in navigation tasks. During our user study, participants often encountered abrupt path changes as the drone found higher reward paths, an emergent phenomenon we dubbed 'switchbacks', a behavior that was overwhelmingly viewed as confusing and unconfident. Participants expressed a preference for more direct paths, desiring an explanation when changes were necessary, echoing previous research findings [2], [10], [34]. By reducing the frequency of these switchbacks, minor losses in expected path optimality may be offset by the associated reduction of workload, freeing up mental capacity for long term planning, as well as improvement in the perceived reliability and trust of the system.

Sudden path changes can be discouraged through the introduction of a small negative reward for path plans that substantially differ from the path of the previous time step, enforcing a degree of plan explicability and predictability [43]. All else being equal, participants also preferred direct arrow paths within the planning horizon, with minimal directional changes, which can also be incentivized through a biasing factor in the reward function. These approaches alleviate high-frequency irritations experienced by human teammates, which can add up over time. Of course, the change in expected reward for a new path may outweigh the desire to maintain predictability, in which case, justification may be necessary.

**A3: Justification.** If an abrupt change in prescriptive guidance is indicated by new observations, it is essential that systems justify this change to avoid losing the trust of their human teammate. Faulty robots are perceived as less trustworthy, but justification has been shown to mitigate the negative impacts of perceived failure [3], [5]. Prior research has shown that robots are perceived as more helpful, intelligent, and trustworthy when they provide the underlying rationale of their decision-making [12], [33]. In addition to these subjective effects, justification offers a snapshot of an agent's inner workings, which users can leverage to gauge the validity of recommended guidance and make well-informed decisions [11].

By leveraging the counterfactual alerts described in A1 and shown in Figure 3 to generate justifications, we can explain to human teammates why they are seeing a sudden change in their recommended path. We posit that tying these alerts to path changes would be preferable to simply triggering the explanations when rewards change. By strategically leveraging system explanation as justification, only utilized during actionable changes in prescriptive guidance within the human's planning horizon, we can avoid saturation of explanation. If explanations are overused, unintended consequences can ensue, increasing workload and leading to habituation, as repeated exposure to explanations reduces user responsiveness to them [17], [18], [21], [37].

## IV. CONCLUSION

In this work, we first synthesized select findings from our prior work [35], which provided design considerations informing AR-based visual communication for human-robot teaming under uncertainty. These findings illustrated the value of providing visual insights into environmental uncertainty alongside robot-generated suggestions, to improve trust, interpretability, performance, and human teammate independence. We used those insights to characterize a set of remaining open challenges for communication in human-robot teaming in uncertain environments, and proposed a set of approaches for addressing them, inspired by techniques in explainable AI.

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
