# OpenReview forum: "Augmented Reality-Based Explainable AI Strategies for Establishing Appropriate Reliance and Trust in Human-Robot Teaming"
_humanrobotinteraction.org/HRI/2022/Workshop/VAM-HRI — VAM-HRI 2022_

### Official Review · Reviewer_o2Wb · 2022-02-25
**Great paper, excited for integrating future approaches**

**Rating:** 9
**Confidence:** 5

**Review:**

This paper addresses the problem of effective communication for human-robot teaming in uncertain environments by leveraging Augmented Reality (AR). Previous work introduced two visual communication modalities for the robot: prescriptive guidance (which visualizes recommended actions, in the form of a spatial arrow) and descriptive guidance (which visualizes the robot’s uncertainty as a heatmap over the shared space). The previous work also introduced MARS, an algorithm for multi-agent collaboration that represents uncertainty with a probability mass function (PMF) and solves two separates Markov Decision Processes (MDPs) to decide what actions the robots should take and how to use the aforementioned communication modalities to suggest actions or express rationale for decisions to the human. Previous work also evaluated MARS by running a within-subjects study that compared conditions where users either were only given 1) prescriptive guidance, 2) descriptive guidance, or 3) a combination of 1) and 2). The main contribution of this work is a synthesis of key takeaways from the user study regarding human-robot collaboration in uncertain settings, and a proposed set of new approaches for addressing open challenges in this problem area.

The main design goals for the new approaches to integrate are 1) leveraging the different visual guidance modalities to help shape human teammates system 1 vs system 2 thinking, and 2) the ability to communicate the quality of the robot’s advice to the human. Based on these, this new work proposes 1) integrating in counterfactual explanations by enabling the agent to use visual and natural language modalities to express reward changes (and confidence in those changes), 2) adding in negative rewards based on how new planned paths differ from previous paths (to reduce cognitive load on user), and 3) adding justifications when robot path plans change to avoid losing human trust.

Overall, these newly identified design goals for future improvement on MARS and proposed future approaches seem very promising and worthwhile to implement and compare against the original MARS system. The paper is very well-written and clear, and I only have possible suggestions for the authors to explore in the future:

1. What was the justification for visualizing the probability mass using color? Is it possible that other means of visualizing the PMF, like showing a cube centered at each spatial location and having a height proportional to the probability mass, would be better? Also, it would be interesting to investigate how to extend the descriptive guidance to a setting with a PDF instead of a discrete range of values like with a PMF.

2. I wonder if it would be useful to enable users to suggest actions to the robot, and enable the human to understand how the robot considers that action under its model. For example, if the human suggests an action to a robot and the robot conveys that the proposed action is bad under it’s model, then the human may choose higher-cognitive ways to diagnose and understand the robot’s model (such as the robot choosing more descriptive guidance than prescriptive when making suggestions). This could be done by evaluating the human’s proposed action or planned trajectory against the policy (or value functions) coming from solving M_{R}

3. Using a negative reward to discourage switch-backs seems like it may be difficult to tune properly, and may prevent the agent from doing the best behaviors possible. Maybe instead, the agent can detect when it is in a switch-base mode (by keeping tracking of how much the path is changing between replans), and if that is happening, instead of showing the planned paths which keep switching back and forth, instead somehow explicitly convey that the robot is considering these different paths right now and is trying to “consider” multiple paths. Essentially, the robot “hides” the low-level details of which plans it is switching between when in this mode, and simply communicates that it is in a state of switching back and forth.

4. It would be interesting to investigate how leveraging human-eye gaze can be used in MARS. It was mentioned that users found it difficult to notice when guidance was outside the field of view, maybe the users eye-gaze can be tracked and used to determine if the humans is  even looking at a guidance feature, which the robot could use to determine whether it needs to take other measures in order to get the users attention.

5. Would it be useful to visualize information about value functions (such as V(s) or Q(s,a)) computed from solving M_r to the human, instead of only communicating the PMF? Maybe the robot could also communicate the advantage value A(s,a) for a particular recommended action to the human, to tell the human how much better they think that action is to the average action under the assumed human’s policy?

---

### Decision · Program_Chairs · 2022-03-04

Accept